# Electrochemical aromatic C–H hydroxylation in continuous flow

Hao Long[1,2,5], Tian-Sheng Chen[1,5], Jinshuai Song [3], Shaobin Zhu [4] & Hai-Chao Xu [1,2✉]

The direct hydroxylation of arene C–H bonds is a highly sought-after transformation but remains an unsolved challenge due to the difficulty in efficient and regioselective C–H oxygenation and high reactivity of the phenolic products leading to overoxidation. Herein we report electrochemical C–H hydroxylation of arenes in continuous flow for the synthesis of phenols. The method is characterized by broad scope (compatible with arenes of diverse electronic properties), mild conditions without any catalysts or chemical oxidants, and excellent scalability as demonstrated by the continuous production of 1 mol (204 grams) of one of the phenol products.

[1] State Key Laboratory of Physical Chemistry of Solid Surfaces, College of Chemistry and Chemical Engineering, Xiamen University, 361005 Xiamen, China. [2] Key Laboratory of Chemical Biology of Fujian Province, Xiamen University, 361005 Xiamen, China. [3] Green Catalysis Center, College of Chemistry, Zhengzhou University, 450001 Zhengzhou, China. [4] NanoFCM INC., Xiamen Pioneering Park for Overseas Chinese Scholars, 361006 Xiamen, China. [5] These authors contributed equally: Hao Long, Tian-Sheng Chen. ✉email: haichao.xu@xmu.edu.cn

Phenols are common structural units in natural products and pharmaceuticals and important synthetic intermediates. Phenol itself is one of the most important bulk chemicals and produced in a multimillion ton scale yearly from benzene through the three-step cumene process[1]. The direct hydroxylation of arene C–H bonds is a straightforward and ideal approach to phenols but remains rather challenging, especially when sustainability is taken into consideration. Established transition metal-catalyzed arene C–H hydroxylation methods usually rely on directing groups to improved reactivity and site-selectivity (Fig. 1a)[2–7]. Non-directed methods have also been reported employing iron catalysis with a special peptide ligand[8] (Fig. 1b, top) or under metal-free conditions with stoichiometric hazardous peroxides as the oxygen source (Fig. 1b, bottom)[9–11]. Oxidant-free conditions are attractive for synthesis because of improved atom economy, reduced cost, and attenuated safety issues associated with the use of stoichiometric chemical oxidants[12]. In this direction, photocatalytic[13] and photoelectrocatalytic[14] methods have been disclosed for the direct hydroxylation of electron-neutral and -deficient arenes (Fig. 1c). In these reactions, a cobalt complex or electrochemistry is employed to facilitate H$_2$ evolution obviating the need for sacrificial chemical oxidants. Limitations of these photochemical methods include incompatibility with electron-rich arenes, the difficulty for scale-up, and the need for UV light or large excess of LiClO$_4$.

Organic electrosynthesis has long been recogwhere as a green synthetic tool and received tremendous renewed interest in recent years[15–30]. A major hurdle for electrochemical C–H hydroxylation is that the phenolic products are much more readily oxidized than the starting arene, leading to overoxidation. However, it has been shown that the electrochemical arene hydroxylation with trifluoroacetic acid (TFA) as the oxygen donor to produce aryl trifluoroacetate intermediates followed by hydrolysis can attenuate the overoxidation issue (Fig. 1d, top)[31–33]. While this approach is promising, the established batch protocols are limited to few examples of electron-neutral and deficient arenes and often require divided cells and highly acidic TFA as the solvent or

cosolvent[31–33]. Electron-rich arenes failed completely[33] or were inefficient even with controlled potential electrolysis[34] because of overoxidation problems, the low reactivity of the intermediate arene radical cations, and self-coupling reactions leading to dimerization and polymerization (the electron-rich arene can compete with the trifluoroacetate as a nucleophile for the arene radical cation intermediate)[33,34].

Continuous-flow electrochemical microreactors are characterized by low residence time, improved mass transfer, and high surface-to-volume ratio[35–47]. These features are helpful to increase reaction efficiency and reduce overoxidation[48]. Very recently, Ošeka and coworkers reported an intriguing electrochemical hydroxylation of electron-rich arenes but are limited to certain electron-rich arenes (Fig. 1d, bottom)[49]. Benzene and electron-deficient arenes failed completely. We have been interested in the application of continuous-flow electrochemistry in organic synthesis[50–54].

In this work, we report electrochemical hydroxylation of arenes of diverse electronic properties (electron-rich, -neutral, and -deficient) in continuous flow for the synthesis of phenols. The high selectivity of our method allows efficient late-stage functionalization of natural products and drug molecules. In addition to its exceptionally broad scope, electrosynthesis requires no catalysts or chemical oxidants and is scalable as demonstrated by the continuous production of 1 mol (204 g) of one of the phenolic products.

## Results

**Reaction development.** 1-Methoxy-4-(trifluoromethyl)benzene **1** was chosen as a model substrate for reaction optimization (Table 1). Under the optimized conditions, the single-pass continuous-flow electrochemical microreactor contained a graphite anode, a Pt cathode, and a spacer with a thickness of 150 μm. A solution of **1** in tBuOMe/MeCN (1:1) containing TFA (2 equiv) and 2,6-lutidine (2 equiv) was pumped at 0.4 mL min$^{-1}$ through the cell (operated at a constant current of 81 mA, 2.8 F mol$^{-1}$) to afford the desired

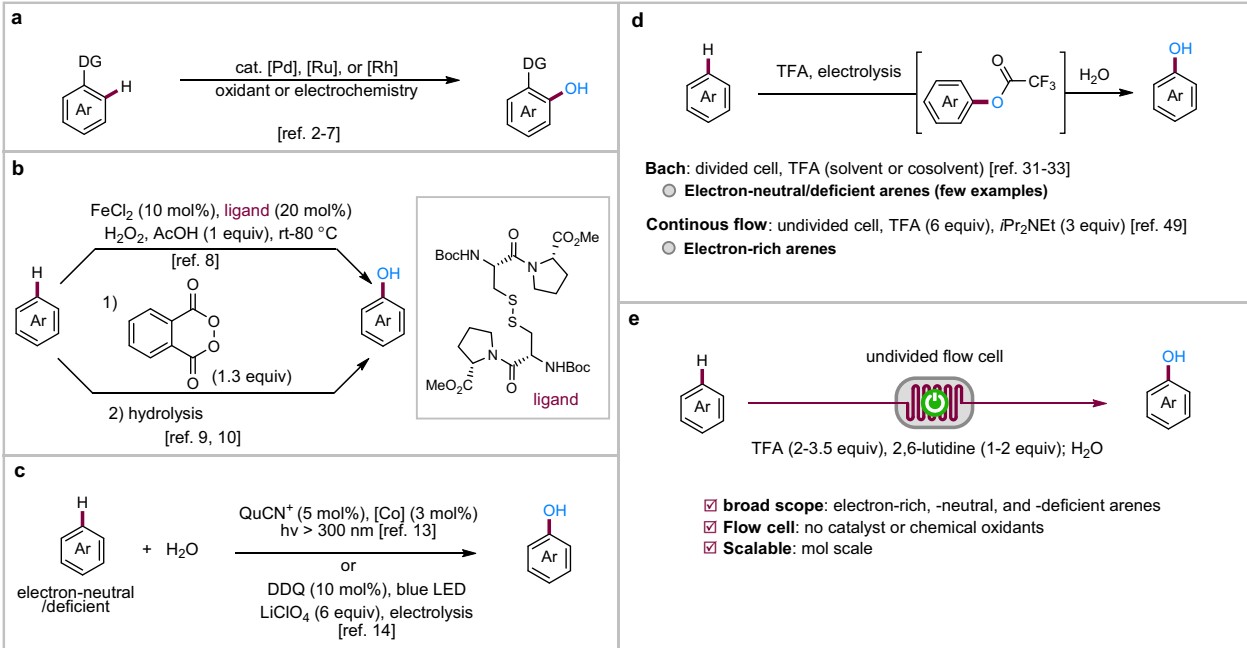

**Fig. 1 C–H hydroxylation of arenes. a** Transition metal-catalyzed C–H hydroxylation with directing groups. **b** C–H hydroxylation employing peroxides. **c** Photocatalytic/electrophotocatalytic hydroxylation of arenes. **d** Electrochemical hydroxylation of arenes. **e** Electrochemical hydroxylation of arenes of in continuous flow (this work). DG directing group, QuCN$^+$ 3-cyano-1-methylquinolinium ion, DDQ 2,3-dichloro-5,6-dicyanobenzoquinone, TFA trifluoroacetic acid.

**Table 1 Optimization of reaction conditions[a].**

| Entry | Deviation from standard conditions | Yield of 2 (%)[b] |
|---|---|---|
| 1 | None | 83 (9) |
| 2 | MeCN as solvent | 78 (5) |
| 3 | tBuOMe as solvent | 8 (38) |
| 4 | Pyridine instead of 2,6-lutidine | 27 (21) |
| 5 | $Et_3N$ instead of 2,6-lutidine | 31 (60) |
| 6 | $iPr_2NEt$ instead of 2,6-lutidine | 23 (60) |
| 7 | $MeCO_2H$ instead of TFA | 0 (34) |
| 8 | TfOH instead of TFA | 0 (36) |
| 9 | $HBF_4$ instead of TFA | 0 (45) |
| 10 | Interelectrode distance = 250 µm | 35 (26) |
| 11[c] | Reaction in batch | 29 (56) |
| 12[d] | Reaction in batch | 15 (28) |

TfOH trifluoromethanesulfonic acid.
[a]Electrolysis conditions: graphite anode, Pt cathode, electrode surface (10 $cm^2$), interelectrode distance = 150 µm, flow rate = 0.40 mL $min^{-1}$, residence time = 22 s (calculated), constant current = 81 mA (2.8 F $mol^{-1}$), **1** (0.045 M). The outlet solution was collected for 5 min (2 mL). [b]Determined by [1]H NMR analysis using 1,3,5-trimethoxybenzene as internal standard. Unreacted **1** is given in brackets. [c]Batch reaction with 0.36 mmol of **1** for 3.4 h (2.8 F $mol^{-1}$), graphite plate anode (1 cm × 1 cm), Pt plate cathode (1 cm × 1 cm). [d]Batch reaction with 0.36 mmol of **1** for 5.0 h (4.2 F $mol^{-1}$), graphite plate anode (1 cm × 1 cm), Pt plate cathode (1 cm × 1 cm).

phenol **2** in 83% yield (entry 1). While the use of pure MeCN as the solvent only caused a slight yield reduction (78% yield, entry 2), reaction in tBuOMe afforded only 8% yield of **2** (entry 3). Subsequent studies on the reaction scope with other substrates revealed that better results were generally obtained by employing the mixed solvent of tBuOMe/MeCN (1:1) than MeCN. Other bases such as pyridine (entry 4), $Et_3N$ (entry 5), or $iPr_2NEt$ (entry 6) were much less effective in promoting the formation of **2**. These bases might compete with trifluoroacetate to react with **1** derived radical cation. In addition, the alkylamines were oxidized more readily than **1** leading to low conversion of **1**. These results demonstrated the importance of 2,6-lutidine as a base, which was sterically hindered and oxidized at relatively high potential. Other acids such as $MeCO_2H$ (entry 7) and TfOH (entry 8) failed completely likely because of competitive oxidation of the acid and overoxidation of the aryl acetate product in the case of $MeCO_2H$ or low nucleophilicity of $TfO^-$. $HBF_4$ was also not a suitable acid due to the lack of an oxygen atom donor (entry 9). The increase of the interelectrode distance to 250 µm resulted in dramatic yield loss (35% yield, entry 10). Conducting the reaction of **1** in a batch reactor led to the formation of **2** in 29% yield with 56% of **1** unreacted after the pass of 2.8 F $mol^{-1}$ of charge in 3.4 h (entry 11). Increasing the reaction time to 5 h to pass more charge (4.2 F $mol^{-1}$) through the reaction solution increased the conversion but resulted in a lower yield of 15% (entry 12), indicating overoxidation of the aryl trifluoroacetate intermediate. These results showcased the advantage of electrosynthesis in continuous flow with a low residence time (<22 s).

**Evaluation of substrate scope.** We next evaluated the substrate scope of the relatively electron-rich arenes (Fig. 2). The electrochemical hydroxylation reaction was compatible with anisoles substituted at the 4-position with functionalities of various electronic properties such as electron-donating OMe (**3**), halogens (Cl, Br; **4** and **5**) and electron-withdrawing COOH (**6**), $CO_2Me$ (**7**), COMe (**8**), $CONEt_2$ (**9**), and CN (**10**). In contrast, 1,3-dimethoxybenzene failed completely and 4-methoxybenzonitrile afforded a low yield of 20% under Ošeka's conditions[49]. Anisole derivatives bearing a substituent at the 2 or 3 position were also suitable substrates but afforded

regioisomers (**11–13**). Pleasingly, most of the regioisomeric phenols in this study were separated by silica gel chromatography. Several trisubstituted benzenes (**14–18**) also participated in the electrochemical hydroxylation reaction. Chroman-4-one (**19**), benzyloxybenzenes (**20** and **21**), and diaryl ethers (**22–25**) were also tolerated. Biphenyls (**26–28**), naphthalenes (**29–31**), 9-bromophenanthrene (**32**), and pyrene (**33**) reacted successfully. The reaction of 1,4-dibromo-2,3-dimethylbenzene ($E_{p/2}$ = 2.13 V vs SCE) afforded the corresponding phenol **34** in 30% yield because of low conversion (46%), which was likely caused by competitive oxidation of 2,6-lutidine and the ether solvent. Benzene, which was oxidized at even higher potential (2.48 V vs SCE), failed to react. Hence these reaction conditions accommodated substrates with oxidation potentials ranging from about 1.3 V to about 2.1 V. Benzoheterocycles such as benzoxazoles (**36–38**), benzimidazoles (**39** and **40**), benzothiazole (**41**), indazole (**42**), and quinolinone (**43**) reacted regioselectively. The regioselectivity of the electrochemical hydroxylation reaction was similar to electrophilic aromatic substitutions and consistent with those observed for the reactions of arene radical cations[55–57].

The electrochemical hydroxylation reaction tolerated a host of common functional groups (Fig. 2), including alkene (**44**), alkyne (**45**), epoxide (**46**), free alcohol (**47**), alkyl tosylate (**48**), alkyl bromide (**49**), and Boc-protected aminoester (**50**). In addition, the method was applicable to the late-stage functionalization of complex nature products and drug molecules (**51–58**).

To accommodate benzene and electron-deficient arenes, the electrolysis conditions were further investigated (Supplementary Table 2). It was found that the electrochemical hydroxylation of these more difficult arenes benefited from the use of a Pt anode under acidic conditions (3.5 equiv of TFA and 1 equiv of 2,6-lutidine) in MeCN. Pt was more resistant to degradation and deactivation than graphite at high potentials. This material is more expansive (about 200 USD for each cell) but is durable and can be reused for years. The acidic conditions helped cathodic $H_2$ evolution to avoid unwanted cathodic reduction of the electron-deficient substrates and intermediates and ensured complete protonation of 2,6-lutidine to reduce its oxidative decomposition. Under these conditions (Fig. 3), the electrochemical method was

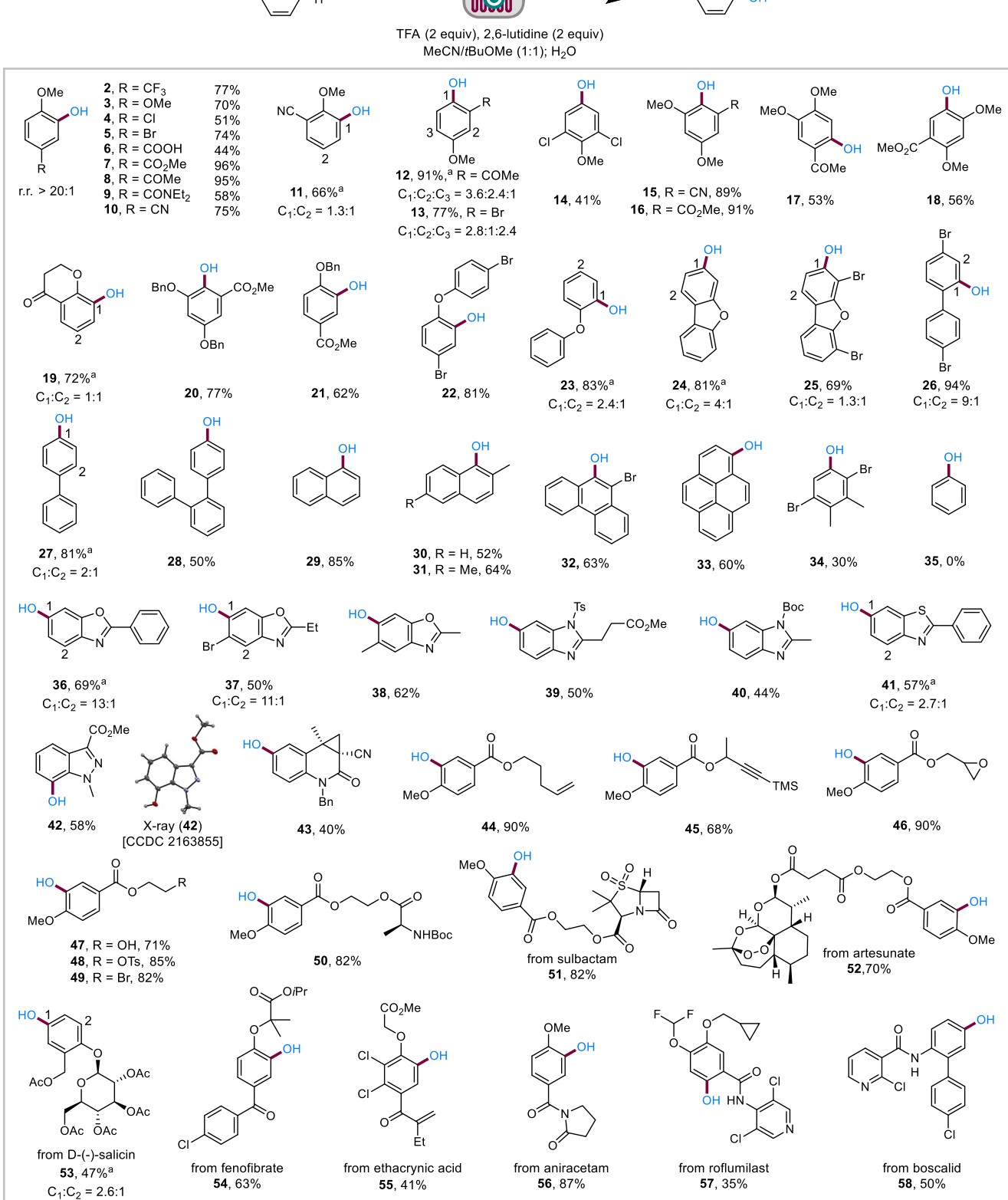

**Fig. 2 Reaction scope of electron-rich arenes.** Regioisomers were not observed unless otherwise mentioned. Reaction conditions: 0.36 mmol of arene, graphite anode, Pt cathode, 52–98 mA, 0.4 mL min⁻¹. ᵃRegioisomers separated by flash chromatography. Ts tosyl, Boc *tert*-butyloxycarbonyl, Bn benzyl, Ac acetyl.

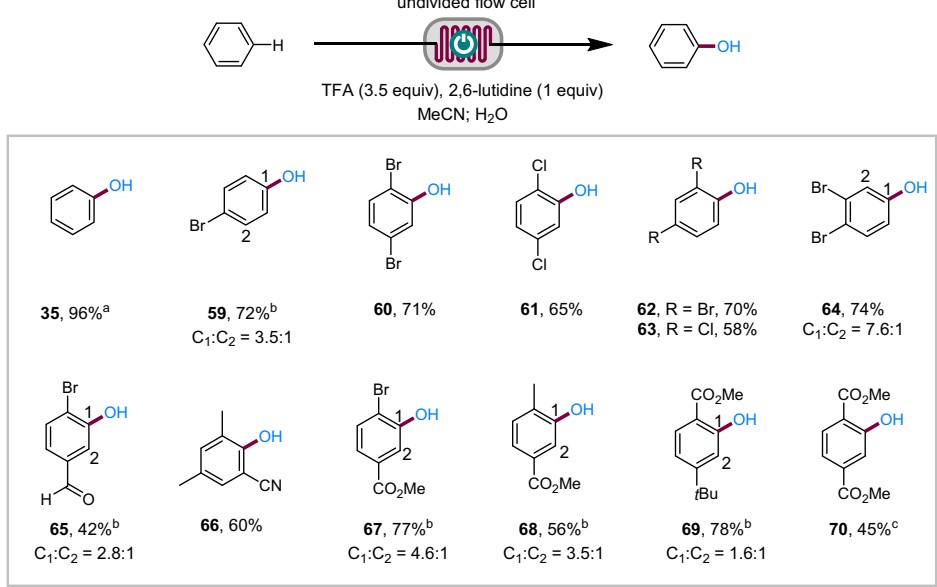

**Fig. 3 Reaction scope of electron-deficient arenes.** Regioisomers were not observed unless otherwise mentioned. Reaction conditions: Pt cathode and anode, constant current = 61–173 mA, 0.3 mL min⁻¹. ªDetermined by GC analysis. ᵇRegioisomers separated by flash chromatography. ᶜReaction in $CH_2Cl_2$ in the presence of 6 equiv of TFA and 2 equiv of 2,6-lutidine.

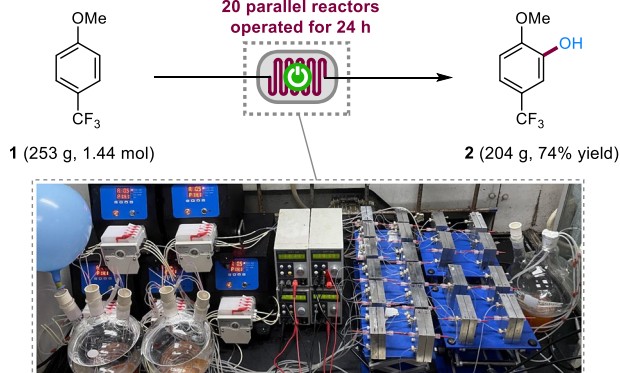

**Fig. 4 Reaction scale-up employing 20 parallel reactors.** Reaction conditions: graphite anode, Pt cathode, MeCN, constant current = 200 mA (for each reactor), 0.5 mL min⁻¹ (for each reactor), **1** (0.1 M), 2.5 F mol⁻¹, rt, 24 h.

applicable to benzene (**35**), mono- and dihalogenated benzenes (**59–64**), 4-bromobenzaldehyde (**65**), 3,5-dimethylbenzonitrile (**66**), and benzoate esters (**67–69**). Dimethyl terephthalate (**70**) with an oxidation potential of 2.70 V ($E_{p/2}$ vs SCE, see the Supplementary Information for the voltammogram) also participated in the electrochemical C–H hydroxylation but required the use of $CH_2Cl_2$ as solvent and higher concentration of TFA (6 equiv) and 2,6-lutidine (2 equiv).

The scale-up of the electrosynthesis of phenol **2** was accomplished through simultaneous scale-out (passing more material through the reactor) and numbering up using 20 parallel reactors (Fig. 4)[58]. A solution of **1** (253 g, 1.44 mol) in MeCN was pumped through the reactors in 24 h to afford **2** in 74% yield (204 g, 1.06 mol). *t*BuOMe was omitted from the solvent to reduce cost and its omission did not affect much of the yield of **2**. The productivity of each reactor was increased by increasing the electric current and flow. These modifications did not affect the reaction efficiency, showcasing the robustness of the method.

**Mechanistic studies**. The regioselectivity observed in our study is consistent with that reported for the reactions of arene radical cations with nucleophiles and can be predicted by the computed singly occupied molecular orbital (SOMO) of the corresponding arene radical cation[55,56]. As shown in Fig. 5a, the distribution of the SOMO of methyl 4-methoxybenzoate derived radical cation on the position ortho to the OMe group is much higher than the meta position, explaining the observed exclusive hydroxylation ortho to the OMe group to give **7**. The SOMOs of 2-phenylbenzoxazole and 2-phenylbenzothioazole are delocalized throughout the skeletons with the distributions on C6 atoms being higher than other H-bearing carbon atoms. Experimentally, these heterocycles reacted mainly to give C6 hydroxylation products.

The oxidation potentials of compounds **1**, **2**, and the corresponding trifluoroacetate **71** were measured in MeCN to be 2.06 V, 1.43 V, 2.32 V ($E_{p/2}$ vs SCE), respectively (Fig. 5b). The much lower oxidation potential of the phenol product **2** than **1** suggested the challenges in aromatic C–H hydroxylation. The increase of oxidation potential upon trifluoroacetoxylation is essential for the success of the electrochemical C–H oxidation. Note that this potential gap did not prevent overoxidation over the hours-long electrolysis in a batch reactor (see Table 1, entries 11 and 12), showcasing the great synthetic potential of continuous-flow electrochemical technology.

A possible mechanism for electrochemical arene hydroxylation is shown in Fig. 5c. Anodic oxidation of the arene generates its corresponding radical cation, which reacts with trifluoroacetate to produce a cyclohexadienyl radical[49,59]. Further one-electron oxidation followed by proton loss generates the aryl trifluoroacetate intermediate, which has been observed experimentally (Supplementary Fig. 4). The aryl trifluoroacetate is hydrolyzed to give the final phenol product upon treating with $H_2O$ during the workup. At the cathode, protons are reduced to $H_2$. Since the electrochemical C–H trifluoroacetoxylation process consumes protons, the concentration of free 2,6-lutidine increases with the increasing conversion of the arene under the reaction conditions with equal amount of TFA and 2,6-lutidine added. Under these conditions, 2,6-lutidine ($E_{p/2}$ = 2.10 V vs SCE) likely serves also as an overcharge protectant[60].

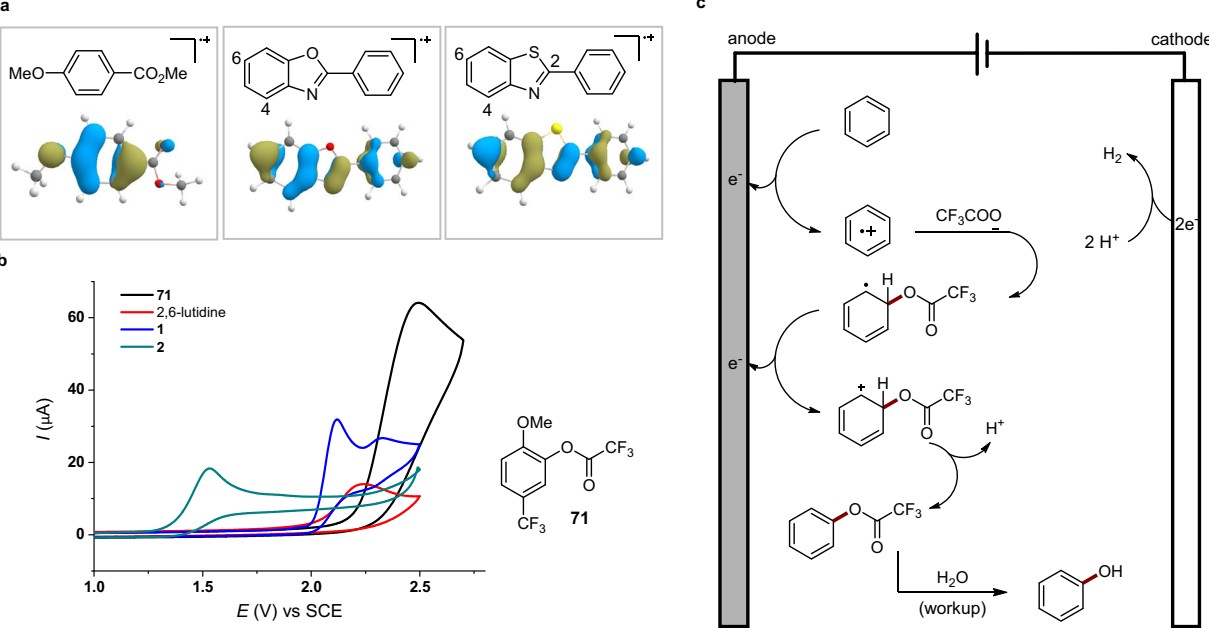

**Fig. 5 Mechanistic studies and proposal. a** Computed SOMO of radical cations at ROM062X/6-31G* level of theory. C, O, S, and H atoms are colored in grey, red, yellow, and white, respectively. The LUMOs are visualized by blue and dark yellow isosurfaces. **b** Cyclic voltammograms obtained in MeCN. **c** Mechanistic proposal.

In summary, we have developed an electrochemical method to achieve C(aryl)–H hydroxylation of arenes with diverse electronic properties without use any catalysts or chemical oxidants. The formation of aryl trifluoroacetate intermediate in a continuous-flow electrochemical microreactor contributes to the success of the C–H hydroxylation process.

## Methods

**Representative procedure for the C–H hydroxylation of electron-rich arenes.** The electrolysis was conducted using a flow electrolytic cell equipped with a graphite anode and a Pt cathode with the exposed surface area of 10 cm$^2$ and interelectrode distance of 150 μm (Supplementary Fig. 1). The solution containing arene substrate (0.045 M), TFA (2.0 equiv), 2,6-lutidine (2.0 equiv) in dry MeCN/tBuOMe (1:1) was pushed using a syringe pump to pass through the flow cell operated with a flow rate of 0.40 mL min$^{-1}$ and a constant current in the range of 52–98 mA. Note that a quick screening of the constant current was performed for each substrate to ensure a maximum yield. The outlet solution was collected for 20 min (8 mL) and quenched with saturated NaHCO$_3$ and extracted with ethyl acetate. To remove 2,6-lutidine, the organic extracts were treated with 2 N HCl, extracted with ethyl acetate, and concentrated under reduced pressure. The residue was chromatographed through silica gel eluting with ethyl acetate/hexanes or methanol/dichloromethane to give the product.

**Representative procedure for C–H hydroxylation of benzene and electron-deficient arenes.** The electrolysis was conducted using a flow electrolytic cell equipped with a Pt anode and a Pt cathode with the exposed surface area of 10 cm$^2$ and interelectrode distance of 150 μm (Supplementary Fig. 2). The solution containing arene substrate (0.06 M), TFA (1.26 mmol, 3.5 equiv), 2,6-lutidine (0.36 mmol, 1.0 equiv) in dry MeCN was pushed using a syringe pump to pass through the flow electrolytic cell operated with a flow rate of 0.30 mL min$^{-1}$ and a constant current (61–173 mA). The outlet solution was collected for 20 min (6 mL). The workup procedure was the same as described above.

## Data availability

The X-ray crystallographic coordinates for structures reported in this study have been deposited at the Cambridge Crystallographic Data Centre (CCDC), under deposition number 2163855. These data can be obtained free of charge from The Cambridge Crystallographic Data Centre via www.ccdc.cam.ac.uk/data_request/cif. The data supporting the findings of this study are available within the article and its Supplementary Information files. Any further relevant data are available from the authors on request.

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

## Acknowledgements

The authors acknowledge NSFC (Nos. 22121001 and 21971213) and Fundamental Research Funds for the Central Universities for support.

## Author contributions

H.L., T.-S.C., J.S. and S.Z. performed the experiments and analyzed the data. H.C.X. designed and directed the project and wrote the manuscript.

## Competing interests

The authors declare no competing interests.
