## [Peer Review File · Nature Communications]

REVIEWER COMMENTS

Reviewer #1 (Remarks to the Author):

In this manuscript the authors have reported the electrochemical aromatic C–H hydroxylation in continuous flow. The authors demonstrate an improvement of an already existing methodology published by Oseka and co-workers where also electron-rich and neutral arenes were electrochemically oxidized and trapped by TFA as oxygen source. The cleavage of this TFA based ester under aqueous conditions give rise of the liberated phenol species. Besides the similarity of those electrochemical flow methodologies the authors were able to establish a very broad substrate scope with even electron-poor derivatives and an impressive up-scaling up to 200 g scale using 20 electrochemical flow cells in a parallel fashion running for 24 h in continuous flow. Nevertheless, since these methodologies are very similar, a direct comparison seems sufficient and reasonable in context of this assessment. The main differences between these two strategies are (1) lutidine use instead of DIPEA (2) platinum electrodes instead of stainless steel (3) stoichiometry of reagents (4) solvent mixture.

(1)+(3) As stated in the main paragraph of the results and discussion the use of lutidine may be advantageous due to the higher oxidation potential in comparison to DIPEA and/or other aliphatic amines. Those amines are commonly applied as terminal reductant in many photoredox and electrochemical reduction procedure because of their low oxidation potential and abundance. The problem is that aliphatic amines would have competed with the arene oxidation and this was obviously an unwanted scenario. However, with a careful reading of the Oseka manuscript, the authors of the manuscript presented here could have circumvented the potential-related problems by slightly increasing the TFA stoichiometry relative to the base, which ensures protonation and a significant increase in oxidation potential.

(2) Platinum electrodes have very unique properties and are very inert materials for organic electrosynthesis. Easy to handle and to clean. Just by the use of a Bunsen-burner. However, as stated in the present manuscript the electrochemical flow reactor is much more expensive especially when electron-poor substrates were tackled due to the use of two platinum electrodes for anode and cathode. In comparison Oseka and co-workers employed inexpensive and also easy to handle stainless steel cathodes. Just rinsing with solvents and polishing with sandpaper these electrodes can be reused many times and as they have demonstrated also in a continuous flow fashion and an upscaling as well. Apart from this feature of handling the different electrodes, platinum has a much lower overpotential to hydrogen evolution relatively to stainless steel [-0.1 V (Pt) vs -0.4 V (SS)]. Thus the reduction of protons towards hydrogen gas is facilitated by the use of platinum. In the supporting information the authors share detailed information about the extent of the screening. There it has been stated that also stainless steel was used. Because of the stoichiometry used for this reaction where no excess of acid was available stainless steel might not function properly

towards hydrogen evolution. Slightly higher TFA concentration could have demonstrated already decent results. Next, the development of a methodology is supposed to be easily reproducible and accessible for the scientific community. Oseka's protocol uses stainless steel which is much more affordable for any researcher who wants to reproduce those results.

(4) The reaction presented uses TBME/MeCN mixtures and has demonstrated that single solvent approaches do not give high yields even though 83% with the mixture relatively to 78% with only acetonitrile is not a significant difference and falls within the deviations of q-NMR analysis. There is no real evidence that the mixture is superior. Surprisingly, THF was not screened by the authors even though Oseka and co-workers used THF for their methodology. Additionally, Oseka stated in their publication the use of MeCN revealed lower yields due to presumably small water contamination. Hence, direct hydroxylation occurs and over-oxidation could not be suppressed anymore even by the use of flow technology due to extremely lowered oxidation potentials. This is a major flaw.

The significance and novelty of the presented methodology is in question. There is no doubt that the upscaling of this reaction is impressive and shows high productivity and it demonstrates the benefit of continuous flow, however upscaling is not the main goal/aim for this journal. Based on the above reasons, I would suggest rejecting the manuscript for a high impact journal like Nature Commun. Instead, I would suggest Org Lett, Chem Asian J, or Communications Chemistry as suitable alternatives.

Some comments:

Results and discussion:

Row 72:

Why did you try HBF₄ anyway? Without oxygen there is no reason to use this as hydroxylation source.

Row 73:

Some reasoning about the interelectrode gap would be interesting. Any potential change maybe?

Row 74-78:

The comparison with a batch reaction is reasonable because this validates the necessity of flow technology. Anyhow your conversion is quite low as you stated at 4.2 F a yield of 29% and RSM of 56%. If you would have run the experiment longer than 4.2 F you could have reached higher conversion. Adding up these numbers is 80+%. Major flaw. Additionally, if 10 cm² is the active

surface area of the flow cell, 81 mA was applied which translates into 2.8 F in the flow cell. That accounts for a current density of 8 mA/cm². In batch you apply 8 mA without any information of the active surface area. So this can only be compared if you apply similar current densities.

Row 112:

Please state that you have measured them or at least make a comment that the CV data is presented in the supporting information.

Supporting Information:

In the section of the scale-up. Please specify your isolation method in more detail. 200 g scale is definitely not a common situation and requires special equipment and techniques. More details on that would be good.

Reviewer #2 (Remarks to the Author):

In this manuscript, Xu and coworkers developed a method for the electrochemical C–H hydroxylation of arenes. This reaction is compatible with a range of electronically diverse arene substrates, is applied to the late stage functionalization of complex molecules, and is readily scaled to produce >200 g of product through parallel flow electrolysis.

A classic challenge in the C–H hydroxylation of arene substrates is that the desired product is more easily reduced than the starting material leading to degradation of the product. To circumvent this issue, previous electrochemical strategies have used TFA as the oxygen source as the resulting trifluoroacetate product has a slightly higher oxidation potential than its requisite starting material. While this strategy provides access to phenol products after a simple aqueous workup, the trifluoroacetate product and arene starting material have close oxidation potentials and therefore solvent quantities of TFA are required to prevent over-oxidation, limiting the functional group compatibility of these previous methods.

Key to the discovery Xu and coworkers report herein was the recognition that continuous flow electrolysis prevents over-oxidation of the desired trifluoroacetate product, allowing for high yields of product without the need for solvent quantities of TFA. A side by side comparison demonstrates

that continuous flow electrolysis significantly outperforms batch electrolysis, forming 83% and 29% product respectively. Not only does flow mitigate over-oxidation, but also reduces the amount of TFA to only 2 equiv, allowing for functional groups like epoxides and tert-butyl carbamates to be tolerated. Due to the synthetic utility of this method, I expect this work to be of interest to the broad readership of Nature Communications and recommend it for publication as is.

Reviewer #3 (Remarks to the Author):

In the manuscript, Xu et al reported an elegant method to achieve selective C–H hydroxylation of arenes in continuous flow without assistance of any directing groups. An exceptionally broad scope of phenolic products with a host of common groups can be furnished from diverse arenes using an undivided cell, even including very challenging benzene and electron-deficient arenes. The development of the reaction conditions is presented in a logical manner, and the choice of the reagents is clearly presented. In general, the chemistry described here is very nice and I am willing to accept it.

1) Aromatic C–H bonds (such as 31, 34, 66) outperformed the benzylic C–H bonds in participation in C–O coupling is very interesting. Is there any benzylic C–O coupling products generated in these reactions?

2) How about the reactivity of aryl iodides in this protocol, is deiodination C–O coupling products formed or other reactions takes place?

We thank the reviewers for taking their valuable time to check the manuscript and for giving constructive suggestions to improve it. We have addressed every comment of the reviewers. A point-to-point response to the comments is as following. The original reviewer comments are in blue.

Reviewer #1 (Remarks to the Author):

In this manuscript the authors have reported the electrochemical aromatic C–H hydroxylation in continuous flow. The authors demonstrate an improvement of an already existing methodology published by Oseka and co-workers where also electron-rich and neutral arenes were electrochemical oxidized and trapped by TFA as oxygen source. The cleavage of this TFA based ester under aqueous conditions give rise of the liberated phenol species. Besides the similarity of those electrochemical flow methodologies the authors were able to establish a very broad substrate scope with even electron-poor derivatives and an impressive up-scaling up to 200 g scale using 20 electrochemical flow cells in a parallel fashion running for 24 h in continuous flow. Nevertheless, since these methodologies are very similar, a direct comparison seems sufficient and reasonable in context of this assessment. The main differences between these two strategies are (1) lutidine use instead of DIPEA (2) platinum electrodes instead of stainless steel (3) stoichiometry of reagents (4) solvent mixture.

Response: We thank the reviewer for pointing out that our method has a very broad substrate scope with even electron-poor derivatives and is impressive in terms of up-scaling. As the reviewer pointed out, the method by Oseka and coworkers worked for electron-rich arenes. The method of Oseka and coworkers failed completely on electron neutral arenes such as benzene and electron-deficient ones. Our conditions are compatible with electron-rich, electron-neutral and electron deficient arenes. Phenol synthesis is an important topic as demonstrated by recent high-profile publications: *Science* **2021**, *372*, 145; *Science* **2021**, *374*, 77-81; *Nature* **2022**, *604*, 677-683. These methods all involve transition metal catalysis with special ligands. The first one requires directing group. The last one need aryl halides as starting materials. We have achieved scalable, broad scope, direct C-H hydroxylation without transition metal catalysts, ligands, and oxidizing agents. We believe without doubt that our work is a breakthrough.

We also want to point out that we and the Oseka group developed the methods independently. Their work came up online on March 9, 2022 and we submitted our work on April 4, 2022. Are our results expected based on the results of Oseka? We don't think so. Oseka and coworkers are experts of this chemistry and would have improved the scope if it is that simple. As the reviewer pointed out, our conditions are very different from those of Oseka including base, solvent, electrode, reagent stoichiometric, not mention the flow conditions and reactor.

(1)+(3) As stated in the main paragraph of the results and discussion the use of lutidine may be advantageous due to the higher oxidation potential in comparison to DIPEA and/or other aliphatic amines. Those amines are commonly applied as terminal reductant in many photoredox and electrochemical reduction procedure because of their low oxidation potential and abundance. The problem is that aliphatic amines would have competed with the arene oxidation and this was obviously an unwanted scenario. However, with a careful reading of the Oseka manuscript, the authors of the manuscript presented here could have circumvented the potential-related problems by slightly increasing the TFA stoichiometry relative to the base, which ensures protonation and a significant increase in oxidation potential.

Response: We thank the reviewer for putting forward this point. If I am not mistaken, the reviewer suggests that the scope of the Oseka work can be expanded by increasing the TFA stoichiometry. There is no evidence that it is the case. Oseka and coworkers are experts of this chemistry. If it is this simple, they would have done it. In fact, the oxidation of the amine is not the only issue here. As I quote from the Supporting Information of the Oseka work: "On the other hand, compounds with oxidation potentials higher than 2.1 V stayed almost intact under the optimized conditions (**B**). In this case, oxidation of THF (as a compound with the lowest oxidation potential) occurred and the corresponding products were observed in GC-MS". They have similar statement in their manuscript: For the substrates with higher E_{ox} only traces of products were observed, while solvent oxidation was prevalent. As you can see, they observed that solvent oxidation is the process in the cases of substrates of higher oxidation potentials. We have measured the voltammogram of 1,4-dibromo-2,3-dimethylbenzene in THF. This compound is oxidized at 2.13 V ($E_{p/2}$ vs SCE in MeCN). As you can see from Figure 1 below, THF starts to get oxidized from around 1.9 V vs SCE and no oxidation peak for 1,4-dibromo-2,3-dimethylbenzene was

observed in THF due to solvent oxidation. This is also what Oseka and coworkers have observed in their studies.

Figure 1. Cyclic voltammogram of 1,4-dibromo-2,3-dimethylbenzene in THF (0.1 M $n\text{Bu}_4\text{NPF}_6$). (A) Background. (B) 1,4-dibromo-2,3-dimethylbenzene (6 mM).

(2) Platinum electrodes have very unique properties and are very inert materials for organic electrochemistry. Easy to handle and to clean. Just by the use of a Bunsen-burner. However, as stated in the present manuscript the electrochemical flow reactor is much more expensive especially when electron-poor substrates were tackled due to the use of two platinum electrodes for anode and cathode. In comparison Oseka and co-workers employed inexpensive and also easy to handle stainless steel cathodes. Just rinsing with solvents and polishing with sandpaper these electrodes can be reused many times and as they have demonstrated also in a continuous flow fashion and an upscaling as well. Apart from this feature of handling the different electrodes, platinum has a much lower overpotential to hydrogen evolution relatively to stainless steel [-0.1 V (Pt) vs -0.4 V (SS)]. Thus the reduction of protons towards hydrogen gas is facilitated by the use of platinum. In the supporting information the authors share detailed information about the extent of the screening. There it has been stated that also stainless steel was used. Because of the stoichiometry used for this reaction where no excess of acid was available stainless steel might not function properly towards hydrogen evolution. Slightly higher TFA concentration could have demonstrated already decent results.

Response: By following the reviewer's suggestion, we have tested the model reaction of compound **1** with a stainless-steel cathode and higher amount of reagents. In this experiment, we employed 6 equiv of TFA and 3 equiv of 2,6-lutidine. The reaction afforded the desired **2** in 47% yield. While the yield is indeed increased, it is still much lower than 83% under our optimal conditions. Plus, the use of acidic conditions reduces functional group tolerance to acid sensitive groups.

Next, the development of a methodology is suppose to be easily reproduceable and accessible for the scientific community. Oseka's protocol uses stainless steel which is much more affordable for any researcher who wants to reproduce those results.

Response: We agree with the reviewer that Pt is much more expensive than stainless steel and good methodology should be easily reproduceable and accessible. As we have already pointed in the manuscript that each Pt electrode for the flow cell costs about 200 USD. If we look at the methodologies in high profile journals, many employs catalysts/ligands or reagents that cost more than 200 USD. In addition, we have been using the same Pt electrodes for years. Hence, the cost of the electrode for one experiment is very little. We don't believe that this amount of cost will prevent researchers from reproducing our results.

More importantly, our method provides access to many phenol products that are not possible with Oseka's protocol.

(4) The reaction presented uses TBME/MeCN mixtures and has demonstrated that single solvent approaches do not give high yields even though 83% with the mixture relatively to 78% with only acetonitrile is not a significant difference and falls within the deviations of q-NMR analysis. There is no real evidence that the mixture is superior. Surprisingly, THF was not screened by the authors even though Oseka and co-workers used THF for their methodology. Additionally, Oseka stated in their publication the use of MeCN revealed lower yields due to presumably small water contamination. Hence, direct hydroxylation occurs and over-oxidation could not be suppressed anymore even by the use of flow technology due to extremely lowered oxidation potentials. This is a major flaw.

Response: We thank the reviewer for raising these points. The reviewer has a good point on the necessity of *t*BuOMe. Although the mixed solvent *t*BuOMe/MeCN is only slightly better than MeCN for the reaction of compound **1**, our later studies of the scope revealed that the mixture of *t*BuOMe/MeCN is generally better than MeCN with other substrates. See below for a few examples.

We have added the following statement to the text.

Subsequent studies on the reaction scope with other substrates revealed that better results were generally obtained by employing the mixed solvent of *t*BuOMe/MeCN (1:1) than MeCN.

We don't know why the reviewer found it surprising that we did not test THF. THF along is not a good solvent for us due to low conductivity and low oxidation potential as we have shown above. Our conditions are better than those of Oseka. To address the reviewer's comment, we have conducted a control experiment with THF or THF/MeCN (1:1) as solvent. With THF alone, the cell potential reached 10 V and increased to 20 V after half an hour operation. In Oseka's case, they employed 6 equiv of TFA and 3 equiv of base, which increase the conductivity. The cell potential remained around 3 V with THF/MeCN (1:1) as solvent. These two experiments gave less than 10% of desired product **2** (Supplementary Table 1, entries 13 and 14).

On the use of MeCN in Oseka's work, their original words are: Potentially, TFA ester is more stable in THF, while in acetonitrile it undergoes hydrolysis leading to the formation of cresol **1**, which is further overoxidized under the electrochemical conditions. As you can see, they used "potentially" because this is all guessing and there is no direct evidence. We thank the reviewer for using "presumably". More importantly, they used 3 equiv of TFA and 2 equiv of Bu₃N, we used different conditions (2 equiv of TFA and 2 equiv of 2,6-lutidine). There are many other factors that can promote the presumed "hydrolysis" such as acid (TFA) or base. There is also no evidence that the presumed formation of cresol is the result of reaction with H₂O. The electrochemical reaction consumes 1 equiv of TFA to form Ar-OCOCF₃. With the consumption of TFA, the nucleophilic Bu₃N or its oxidatively

dealkylated amine product may react with TFA ester intermediate to form cresol. 2,6-Lutidine is much less basic than alkylamine and more hindered and allow us to use 2 equiv of TFA. Our results in Supplementary Table 1 (entry 11) showed that the increase of TFA to 3 equiv resulted in reduced conversion. As you can see from Oseka's results in their paper Table 1, decrease of residence time also helps. For their reaction in MeCN, residence time is 5 min. Under our conditions, residence time is less than 22 seconds. We think the Oseka work is very nice as we have described their work in our manuscript as intriguing. But it is confusing for us why using Oseka's work/words as some sort of "golden standard/rule" to judge our results. We are the ones that have the much better reaction conditions/results.

The significance and novelty of the presented methodology is in question. There is no doubt that the upscaling of this reaction is impressive and shows high productivity and it demonstrates the benefit of continuous flow, however upscaling is not the main goal/aim for this journal. Based on the above reasons, I would suggest rejecting the manuscript for a high impact journal like Nature Commun. Instead, I would suggest Org Lett, Chem Asian J, or Communications Chemistry as suitable alternatives.

Response: We respectively disagree with the reviewer. Phenol synthesis is an important topic as demonstrated by recent high-profile publications: *Science* **2021**, 372, 145; *Science* **2021**, 374, 77-81; *Nature* **2022**, 604, 677-683. These methods all involve transition metal catalysis with special ligands. The first one requires a directing group. The first and second one requires O₂ or H₂O₂ as oxidizing agents. The last one need aryl halides as starting materials. We have achieved scalable and broad scope phenol synthesis via electrochemical C-H hydroxylation without transition metal catalysts, ligands, and oxidizing agents. Many of our substrates are currently not possible with existing methods, including that of Oseka. We believe without doubt our work represents is a breakthrough in phenol synthesis and is significant and novel. The other two reviewers agree with us.

Some comments:

Results and discussion:

Row 72:

Why did you try HBF₄ anyway? Without oxygen there is no reason to use this as hydroxylation source.

Response: We thank the reviewer for raising this good point. Our reactions are conducted under air for convenience (the results are the same under argon). We tested HBF₄ to show TFA is the oxygen source not just an acid, and trace water or O₂ are not the oxygen source.

Row 73:

Some reasoning about the interelectrode gap would be interesting. Any potential change maybe?

Response: We thank the reviewer for raising this good point. We don't have an exact explanation for this observation. The cell potential increased from 3.8 V (0.15 mm gap) to 4.7 V (0.25 mm gap). For this control experiment with 0.25 mm gap, other conditions remained the same. But the residence time increased from 22 seconds to 37.5 seconds. The larger gap also decreases the efficiency of mass transfer (assuming laminar flow and diffusion as the major means of mass transfer), which may have led to the lower conversion and increased overoxidation.

Row 74-78:

The comparison with a batch reaction is reasonable because this validates the necessity of flow technology. Anyhow your conversion is quite low as you stated at 4.2 F a yield of 29% and RSM of 56%. If you would have run the experiment longer than 4.2 F you could have reached higher conversion. Adding up these numbers is 80+%. Major flaw. Additionally, if 10 cm² is the active surface area of the flow cell, 81 mA was applied which translates into 2.8 F in the flow cell. That accounts for a current density of 8 mA/cm². In batch you apply 8 mA without any information of the active surface area. So this can only be compared if you apply similar current densities.

Response: We thank the reviewer for raising these good points. There is misunderstanding of the data here. For the batch reactions, we conducted two experiments (Table 1, entries 11 and 12) for 3.4 h (2.8 F/mol) and 5 h (4.2 F/mol), respectively. The reaction in entry 11 for 3.4 h give 29% yield and RSM of 56%. Reaction for 5 h (entry 12) led to 15% yield and RSM of 28%. These results showed that the increase

of the conversion also led to decomposition of the product because of the close oxidation potential of the intermediate Ar-OCOCF₃ with the starting arene. This observation is consistent with those of Oseka and coworkers using batch reactors. They stated that “Prolonged reaction time causes the decomposition of electron-rich compounds by overoxidation under the electrochemical conditions”. The overoxidation is a significant challenge for C-H oxygenation.

We have included the size of the electrode (1 cm x 1 cm) in the manuscript. For the batch reactors, the calculated current density is about 8 mA/cm², similar to the flow conditions. But we do want to point out that the local current density on the electrode of the flow reactor is not even. In a flow electrochemical cell, the conversion increase from entry to exit (See Figure below, reproduced from R. A. Green, R. C. D. Brown, D. Pletcher, *J. Flow Chem.* **2015**, 5, 31-36). Hence, the local current density will drop along the channel from inlet to outlet (*Chem. Rev.* **2018**, 118, 4573-4591).

Row 112:

Please state that you have measured them or at least make a comment that the CV data is presented in the supporting information.

Response: We thank the reviewer for this suggestion. We have directed the readers to the Supplementary Information for the voltammogram.

Dimethyl terephthalate (**70**) with an oxidation potential of 2.70 V ($E_{p/2}$ vs SCE, see the Supplementary Information for the voltammogram) also participated in the electrochemical C–H

hydroxylation but required the use of CH_2Cl_2 as solvent and higher concentration of TFA (6 equiv) and 2,6-lutidine (2 equiv).

Supporting Information:

In the section of the scale-up. Please specify your isolation method in more detail. 200 g scale is definitely not a common situation and requires special equipment and techniques. More details on that would be good.

Response: We thank the reviewer for this suggestion. We have provided more details on the scale up in the SI. For large scale reaction, the difficult part is isolation of the product. We did not use any special equipment and techniques. We simply divided the crude product to 8 parts with 20-30 grams each and run column chromatography with each part.

Reviewer #2 (Remarks to the Author):

In this manuscript, Xu and coworkers developed a method for the electrochemical C–H hydroxylation of arenes. This reaction is compatible with a range of electronically diverse arene substrates, is applied to the late stage functionalization of complex molecules, and is readily scaled to produce >200 g of product through parallel flow electrolysis.

A classic challenge in the C–H hydroxylation of arene substrates is that the desired product is more easily reduced than the starting material leading to degradation of the product. To circumvent this issue, previous electrochemical strategies have used TFA as the oxygen source as the resulting trifluoroacetate product has a slightly higher oxidation potential than its requisite starting material. While this strategy provides access to phenol products after a simple aqueous workup, the trifluoroacetate product and arene starting material have close oxidation potentials and therefore solvent quantities of TFA are required to prevent over-oxidation, limiting the functional group compatibility of these previous methods.

Key to the discovery Xu and coworkers report herein was the recognition that continuous flow electrolysis prevents over-oxidation of the desired trifluoroacetate product, allowing for high yields of product without the need for solvent quantities of TFA. A side by side comparison demonstrates that continuous flow electrolysis significantly outperforms batch

electrolysis, forming 83% and 29% product respectively. Not only does flow mitigate over-oxidation, but also reduces the amount of TFA to only 2 equiv, allowing for functional groups like epoxides and tert-butyl carbamates to be tolerated. Due to the synthetic utility of this method, I expect this work to be of interest to the broad readership of Nature Communications and recommend it for publication as is.

Response: We thank the reviewer for the positive recommendation.

Reviewer #3 (Remarks to the Author):

In the manuscript, Xu et al reported an elegant method to achieve selective C–H hydroxylation of arenes in continuous flow without assistance of any directing groups. An exceptionally broad scope of phenolic products with a host of common groups can be furnished from diverse arenes using an undivided cell, even including very challenging benzene and electron-deficient arenes. The development of the reaction conditions is presented in a logical manner, and the choice of the reagents is clearly presented. In general, the chemistry described here is very nice and I am willing to accept it.

Response: We thank the reviewer for the positive recommendation.

1) Aromatic C–H bonds (such as 31, 34, 66) outperformed the benzylic C–H bonds in participation in C–O coupling is very interesting. Is there any benzylic C–O coupling products generated in these reactions?

Response: We thank the reviewer for this very good point. For compounds such as 31, 34, 66, we did not observe benzylic oxygenation products under these conditions. Since benzylic oxygenation to alcohols are still a difficult transformation because of overoxidation to ketones, we have been working on optimizing the electrochemical conditions to divert the reaction from ring oxygenation to benzylic oxygenation. We now have good conditions to achieve benzylic oxygenation, which will be reported separately in due course.

2) How about the reactivity of aryl iodides in this protocol, is deiodination C–O coupling products formed or other reactions takes place?

Response: We thank the reviewer for this very good question. We have now tested 4-iodoanisole under the standard conditions. The reaction returned most of the starting material (70%) without observation deiodination products or hydroxylation product. Since aryl iodide is known to be oxidized electrochemically to I(III) compounds, we suspect that 4-iodoanisole is oxidized to I(III) on the anode, which is then reduced back at the cathode to return to the initial aryl iodide, leading to observed low conversion.

I have read the feedback of the authors with great care and answered on the feedback as shown below in black. The answers of the authors have been put in a blue color.

Despite the feedback from the authors, I remain with my standpoint that the Eur J Org Chem paper of Oseka undermines the novelty of this manuscript, making it more suitable for a journal like Organic Letters or Chemistry A European Journal.

We thank the reviewer for putting forward this point. If I am not mistaken, the reviewer suggests that the scope of the Oseka work can be expanded by increasing the TFA stoichiometry.

Referee Comment: That wasn't the point. It was just stated that ensuring protonation of the base circumvent the oxidation of the amine, hence their desired reaction path was possible.

There is no evidence that it is the case. Oseka and coworkers are experts of this chemistry. If it is this simple, they would have done it. In fact, the oxidation of the amine is not the only issue here. As I quote from the Supporting Information of the Oseka work: "On the other hand, compounds with oxidation potentials higher than 2.1 V stayed almost intact under the optimized conditions (B). In this case, oxidation of THF (as a compound with the lowest oxidation potential) occurred and the corresponding products were observed in GC-MS". They have similar statement in their manuscript: For the substrates with higher EOX only traces of products were observed, while solvent oxidation was prevalent. As you can see, they observed that solvent oxidation is the process in the cases of substrates of higher oxidation potentials. We have measured the voltammogram of 1,4-dibromo-2,3-dimethylbenzene in THF. This compound is oxidized at 2.13 V (Ep/2 vs SCE in MeCN). As you can see from Figure 1 below, THF starts to get oxidized from around 1.9 V vs SCE and no oxidation peak for 1,4-dibromo-2,3-dimethylbenzene was observed in THF due to solvent oxidation. This is also what Oseka and coworkers have observed in their studies.

Referee comment: I agree on that. The potential window is important and needs to be considered. As you have clearly demonstrated that the potential window of THF is the limiting factor of Oseka's reaction methodology.

By following the reviewer's suggestion, we have tested the model reaction of compound **1** with a stainless-steel cathode and higher amount of reagents. In this experiment, we employed 6 equiv of TFA and 3 equiv of 2,6-lutidine. The reaction afforded the desired **2** in 47% yield. While the yield is indeed increased, it is still much lower than 83% under our optimal conditions. Plus, the use of acidic conditions reduces functional group tolerance to acid sensitive groups.

Referee comment: I agree on this. The yield does not increase significantly and additionally you are right that higher acid loading can result in lower functional group tolerance.

We agree with the reviewer that Pt is much more expensive than stainless steel and good methodology should be easily reproducible and accessible. As we have already pointed in the manuscript that each Pt electrode for the flow cell costs about 200 USD. If we look at the methodologies in high profile journals, many employ catalysts/ligands or reagents that cost more than 200 USD. In addition, we have been using the same Pt electrodes for years. Hence, the cost of the electrode for one experiment is very little. We don't believe that this amount of cost will prevent researchers from reproducing our results. More importantly, our method provides access to many phenol products that are not possible with Oseka's protocol.

We don't know why the reviewer found it surprising that we did not test THF. THF alone is not a good solvent for us due to low conductivity and low oxidation potential as we have shown above. Our conditions are better than those of Oseka. To address the reviewer's comment, we have conducted a control experiment with THF or THF/MeCN (1:1) as solvent. With THF alone, the cell potential reached 10 V and increased to 20 V after half an hour operation. In Oseka's case, they employed 6 equiv of TFA and 3 equiv of base, which increase the conductivity. The cell potential remained around 3 V with THF/MeCN (1:1) as solvent. These two experiments gave less than 10% of desired product **2** (Supplementary Table 1, entries 13 and 14).

Referee Comment: It was surprising because it has been published recently. That's why I was wondering why you haven't tried that under your conditions. Nevertheless, your explanation is reasonable and makes sense. Plus additional data revealed the narrow electrochemical window of THF, which could be seen as problematic for the conversion of some electron-neutral and electron-poor substrates

On the use of MeCN in Oseka's work, their original words are: Potentially, TFA ester is more stable in THF, while in acetonitrile it undergoes hydrolysis leading to the formation of cresol **1**, which is further overoxidized under the electrochemical conditions. As you can see, they used "potentially" because this is all guessing and there is no direct evidence. We thank the reviewer for using "presumably". More importantly, they used 3 equiv of TFA and 2 equiv of Bu₃N, we used different conditions (2 equiv of TFA and 2 equiv of 2,6-lutidine). There are many other factors that can promote the presumed "hydrolysis" such as acid (TFA) or base. There is also no evidence that the presumed formation of cresol is the result of reaction with H₂O. The electrochemical reaction consumes 1 equiv of TFA to form Ar-OCOCF₃. With the consumption of TFA, the nucleophilic Bu₃N or its oxidatively dealkylated amine product may react with TFA ester intermediate to form cresol. 2,6-Lutidine is much less basic than alkylamine and more hindered and allow us to use 2 equiv of TFA. Our results in Supplementary Table 1 (entry 11) showed that the increase of TFA to 3 equiv resulted in reduced conversion. As you can see from Oseka's results in their paper Table 1, decrease of residence time also helps. For their reaction in MeCN, residence time is 5 min. Under our conditions, residence time is less than 22 seconds. We think the Oseka work is very nice as we have described their work in our manuscript as intriguing. But it is confusing for us why using Oseka's work/words as some sort of "golden standard/rule" to judge our results. We are the ones that have the much better reaction conditions/results.

New comment: I agree that the hydrolysis and/or side reactions in Oseka's case could have been introduced due to other reaction modi or water sources. Nevertheless they have observed this only by the change of the solvent. Their assumption that it could have happened due to water content is reasonable.

We respectively disagree with the reviewer. Phenol synthesis is an important topic as demonstrated by recent high-profile publications: Science 2021, 372, 145; Science 2021, 374, 77-81; Nature 2022, 604, 677-683. These methods all involve transition metal catalysis with special ligands. The first one requires a directing group. The first and second one requires O₂ or H₂O₂ as oxidizing agents. The last one need aryl halides as starting materials. We have achieved scalable and broad scope phenol synthesis via electrochemical C-H hydroxylation without transition metal catalysts, ligands, and oxidizing agents. Many of our substrates are currently not possible with existing methods, including that of Oseka. We believe without doubt our work represents is a breakthrough in phenol synthesis and is significant and novel. The other two reviewers agree with us.

New comment: The novelty of the presented work is not in question because of the importance of the synthesis of phenolic derivatives, which you have shown by citing Science and Nature articles, but because the use of electroflow chemistry as an activation mode and technological niche is not new. On the one hand, the fruitful combination of electrochemistry and flow techniques is a powerful strategy for bypassing side reactions and increasing productivity in the production of oxygenated intermediates, and on the other hand, it emphasizes green aspects and sustainability, especially with respect to previous protocols. As you have noted, those methodologies required external oxidants and transition metal catalysts. BUT this was done by Oseka and co-workers in the journal EurJOC with slightly different conditions. The submitted manuscript is indeed intriguing and important to the scientific community with the broad substrate scope and parallel flow reactor synthesis at 200 g scale, but not appropriate for Nature Communications, a high impact journal. Rejection is still recommended and submission in a journal with lower impact factor suggested.

We thank the reviewer for raising these good points. There is misunderstanding of the data here. For the batch reactions, we conducted two experiments (Table 1, entries 11 and 12) for 3.4 h (2.8 F/mol) and 5 h (4.2 F/mol), respectively. The reaction in entry 11 for 3.4 h give 29% yield and RSM of 56%. Reaction for 5 h (entry 12) led to 15% yield and RSM of 28%. These results showed that the increase of the conversion also led to decomposition of the product because of the close oxidation potential of the intermediate Ar-OCOCF₃ with the starting arene. This observation is consistent with those of Oseka and coworkers using batch reactors. They stated that "Prolonged reaction time causes the decomposition of electron-rich compounds by overoxidation under the electrochemical conditions". The overoxidation is a significant challenge for C-H oxygenation. We have included the size of the electrode (1 cm x 1 cm) in the manuscript. For the batch reactors, the calculated current density is about 8 mA/cm², similar to the flow conditions. But we do want to point out that the local current density on the electrode of the flow reactor is not even. In a flow electrochemical cell, the conversion increase from entry to exit (See Figure blow, reproduced from R. A. Green, R. C. D. Brown, D. Pletcher, J. Flow Chem. 2015, 5, 31-36). Hence, the local current density will drop along the channel from inlet to outlet (Chem. Rev. 2018, 118, 4573-4591).

Referee comment: Ok I misread the data, that's true. Then it makes more sense and the reasoning to use flow chemistry instead of batch chemistry makes also much more sense. Indeed the current densities in flow electro chemistry are not an easy parameter to evaluate because it depends and mainly distributed at the entrance of the flow reactor.

Reviewer #3 (Remarks to the Author):

The authors have responded to the comments point by point very clearly, and I recommend it for publication as is.

A point-to-point response to the comments of reviewers is as following. The original reviewer comments are in black, and our responses are in blue.

Reviewer #1 (Remarks to the Author):

I have read the feedback of the authors with great care and answered on the feedback as shown below in black. The answers of the authors have been put in a blue color. Despite the feedback from the authors, I remain with my standpoint that the Eur J Org Chem paper of Oseka undermines the novelty of this manuscript, making it more suitable for a journal like Organic Letters or Chemistry A European Journal.

Response: We thank the reviewer for taking time to review the manuscript again. The great method by Oseka and coworkers failed with electron neutral arenes such as benzene and electron-deficient ones. We have been able to address this limitation and shown that flow electrochemistry can be compatible with electron-rich, electron-neutral and electron deficient arenes. The topic is highly important, and we have been able to do what others cannot. From this point of view, we believe without doubt that our work is a breakthrough.

That wasn't the point. Its was just stated that ensuring protonation of the base circumvent the oxidation of the amine, hence their desired reaction path was possible.

Response: We thank the reviewer for clarifying this point.

I agree on that. The potential window is important and needs to be considered. As you have clearly demonstrated that the potential window of THF is the limiting factor of Oseka's reaction methodology.

Response: We thank the reviewer for agreeing to our point of view.

I agree on this. The yield does not increase significantly and additionally you are right that higher acid loading can result in lower functional group tolerance.

Response: We thank the reviewer for agreeing to our point of view.

It was surprising because it has been published recently. That's why I was wondering why you haven't tried that under your conditions. Nevertheless, your explanation is reasonable

and makes sense. Plus additional data revealed the narrow electrochemical window of THF, which could be seen as problematic for the conversion of some electron-neutral and electron poor substrates

Response: We thank the reviewer for agreeing to our explanation.

I agree that the hydrolysis and/or side reactions in Oseka's case could have been introduced due to other reaction modi or water sources. Nevertheless they have observed this only by the change of the solvent. Their assumption that is could have happened due to water content is reasonable.

Response: We thank the reviewer for agreeing to our explanation.

The novelty of the presented work is not in question because of the importance of the synthesis of phenolic derivatives, which you have shown by citing Science and Nature articles, but because the use of electroflow chemistry as an activation mode and technological niche is not new. On the one hand, the fruitful combination of electrochemistry and flow techniques is a powerful strategy for bypassing side reactions and increasing productivity in the production of oxygenated intermediates, and on the other hand, it emphasizes green aspects and sustainability, especially with respect to previous protocols. As you have noted, those methodologies required external oxidants and transition metal catalysts. BUT this was done by Oseka and co-workers in the journal EurJOC with slightly different conditions. The submitted manuscript is indeed intriguing and important to the scientific community with the broad substrate scope and parallel flow reactor synthesis at 200 g scale, but not appropriate for Nature Communications, a high impact journal. Rejection is still recommended and submission in a journal with lower impact factor suggested.

Response: We respectfully disagree with the reviewer on the suitability of our work for a high-profile journal. We thank the reviewer for agreeing with us that our work is novel and important. The respected work of Oseka and coworkers only highlights the challenges with electrochemical hydroxylation. We have addressed the challenge of an important topic and achieved what is not possible by other methods. In our view, our work is well-suited for Nature Communications.

Reviewer #3 (Remarks to the Author):

The authors have reponded to the comments point by point very clearly, and I recommend it for publication as is.

Response: We thank the reviewer for taking time to review the manuscript again and for the positive recommendation.